# Characterization of BV6-Induced Sensitization to the NK Cell Killing of Pediatric Rhabdomyosarcoma Spheroids

**DOI:** 10.3390/cells12060906

**Published:** 2023-03-15

**Authors:** Vinzenz Särchen, Lisa Marie Reindl, Sara Wiedemann, Senthan Shanmugalingam, Thomas Bukur, Julia Becker, Martin Suchan, Evelyn Ullrich, Meike Vogler

**Affiliations:** 1Institute for Experimental Cancer Research in Pediatrics, Goethe-University, 60528 Frankfurt, Germany; 2Department for Children and Adolescents Medicine, University Hospital Frankfurt, Goethe-University, 60590 Frankfurt, Germany; 3Experimental Immunology, Goethe-University, 60590 Frankfurt, Germany; 4TRON gGmbH—Translational Oncology, The University Medical Center, The Johannes Gutenberg-University Mainz, 55131 Mainz, Germany; 5German Cancer Consortium (DKTK), Partner Site Mainz/Frankfurt, 69120 Heidelberg, Germany; 6University Cancer Center Frankfurt (UCT), University Hospital Frankfurt, Goethe-University Frankfurt, 60590 Frankfurt, Germany; 7Frankfurt Cancer Institute, Goethe-University, 60590 Frankfurt, Germany

**Keywords:** NK cells, Smac mimetic, BV6, cell death, rhabdomyosarcoma, tumor spheroids

## Abstract

Although the overall survival in pediatric rhabdomyosarcoma (RMS) has increased over the last decades, the most aggressive subtype of alveolar RMS is in dire need of novel treatment strategies. RMS cells evade cell death induction and immune control by increasing the expression of inhibitors of apoptosis proteins (IAPs), which can be exploited and targeted with stimulation with Smac mimetics. Here, we used the Smac mimetic BV6 to re-sensitize RMS spheroids to cell death, which increased killing induced by natural killer (NK) cells. Single BV6 treatment of RMS spheroids did not reduce spheroidal growth. However, we observed significant spheroidal decomposition upon BV6 pre-treatment combined with NK cell co-cultivation. Molecularly, IAPs s are rapidly degraded by BV6, which activates NF-κB signal transduction pathways in RMS spheroids. RNA sequencing analysis validated NF-κB activation and identified a plethora of BV6-regulated genes. Additionally, BV6 released caspases from IAP-mediated inhibition. Here, caspase-8 might play a major role, as knockdown experiments resulted in decreased NK cell-mediated attack. Taken together, we improved the understanding of the BV6 mechanism of RMS spheroid sensitization to cytotoxic immune cells, which could be suitable for the development of novel combinatory cellular immunotherapy with Smac mimetics.

## 1. Introduction

With an incidence of 4–5% of all pediatric cancers, rhabdomyosarcoma (RMS) is the most common form of soft-tissue sarcoma [1]. There are two major subgroups of RMS, the embryonal, or fusion-negative, RMS (eRMS or FNRMS) and the alveolar, or fusion-positive, RMS (aRMS or FPRMS) subtypes [2]. The fusion status refers to the translocation of the genes PAX3/7 to FOXO1, which results in a highly transcriptionally active, fusion protein-driving malignant transformation [3]. Therapeutic advances and multi-modal treatment strategies have increased the overall survival rate of patients with RMS [4,5]. However, the survival rate decreases to 46% for the aRMS subgroup and further to 21% for metastasizing cancers [6].

Cancer cells have developed numerous strategies to escape immune surveillance, as highlighted by the discoveries of checkpoint inhibitors, for which the Nobel Prize was awarded to J.P. Allison and T. Honjo in 2018 [7]. In RMS, the tumor microenvironment (TME) is described as immunosuppressive with a low number of lymphocytes infiltrating in the RMS tumor mass. NK cells were among the investigated infiltrating lymphocytes and can directly induce a cytolytic effect upon target cell recognition [8,9]. Cell death in target cells can be induced by the secretion of cytotoxic granules containing perforin and granzymes. Upon perforin-dependent entry of granzymes into the target cells, granzymes cleave their target proteins and can prompt mitochondria to induce the intrinsic apoptotic machinery [10,11]. After the secretion and depletion of cytotoxic granules, NK cells can switch to a death receptor (i.e., Fas/FasL, TRAIL/TRAIL-R1/2 and TNF-R1/TNFα)-dependent killing mechanism. Upon ligation of the respective death receptor, intracellular target proteins (FADD, TRADD and pro-caspase-8) are recruited to form a death domain-dependent death-inducing signaling complex (DISC), which leads to the cleavage of caspase-8, and the activation of the caspase cascade and extrinsic apoptotic machinery [12,13]. The role of caspases in the programmed cell death pathway was initially described in the model organism *C. elegans*, for which the Nobel Prize was jointly awarded to S. Brenner, H. R. Horvitz and J. E. Sulston in 2002 [14].

One approach to increase such cytotoxic immune cell attack by NK cells could be to follow the strategy for which the Nobel Prize was awarded in 2018, that is, the inhibition of negative immune regulators [7]. Here, we combined a cellular immunotherapeutic approach using NK cells with the fundamental idea for which the Nobel prize was awarded to G. Eliot and G. Hitchings in 1988, that is, to utilize a specific cancer targeting agent [15]. We used the Smac mimetic BV6 as a specific inhibitor of apoptosis protein (IAP) targeting agent, which mimics the endogenous protein Smac and leads to the degradation of IAP proteins, e.g., XIAP, cIAP1 and cIAP2 [16,17]. RMS cells evade cell death by up-regulating cIAP1 [18] and depend on XIAP expression for survival [19]; hence, antagonizing either cIAP1 or XIAP may result in increased sensitivity to apoptotic stimuli. Besides by directly binding to and inhibiting caspases, IAPs are involved in a variety of cellular functions, e.g., NF-κB signal transduction. cIAP1/2 are associated with the tumor necrosis factor receptor 1 (TNFR1) complex and regulate the interplay of the canonical and non-canonical NF-κB signaling pathways [20,21,22].

Previous studies performed in our lab showed that treatment with cell death-sensitizing agents, e.g., BH3 [23] or Smac mimetics [24,25], induces immune cell vulnerability. Here, we used the Smac mimetic BV6 as a cell death-inducing agent in combination with NK cells to trigger an increased killing effect on RMS spheroids. Cytotoxic NK cells can recognize and kill an eRMS/FNRMS cell line (RD) to a certain degree, but they fail to fully recognize an aRMS/FPRMS cell line (RH30) as a target. Fischer et al. initially established a BV6-mediated effect of IL-2 activated expanded NK cell attack sensitization on two-dimensionally (2D) cultured RMS cells. The putative involvement of the TNF-related apoptosis-inducing ligand (TRAIL) receptor system and transcriptional up-regulation of the TRAIL receptors (DR4/5) could be shown [24]. However, the exact molecular mechanisms driving the sensitizing effect have not yet been investigated in further detail. Based on this study, we utilized 3D-cultured RMS spheroids to establish a relevant involvement of the BV6 sensitizing effect in an in vivo-like cell culture model. Further, an exploratory RNA sequencing approach revealed underlying BV6-induced signaling pathways that putatively drive the immune cell sensitization of RMS spheroids. Here, the NF-κB signaling pathway, in addition to the NK cell killing caspase dependency of the targeted RMS spheroids, is analyzed in detail.

## 2. Materials and Methods

### 2.1. Materials

The sensitizing effect was achieved with pre-treatment for 24 h with the bivalent Smac mimetic BV6 (kindly provided by Genentech). Cell death was induced with treatment with human recombinant TRAIL (R&D Systems, Wiesbaden, Germany). Cell death was inhibited with zVAD.fmk (Bachem, Heidelberg, Germany), anti-TRAIL antibody (No. 2E5; Enzo Life Sciences, Lörrach, Germany), zAAD-CMK (Calbiochem Merck, Darmstadt, Germany) and Etanercept (Enbrel, Merck, Darmstadt, Germany).

### 2.2. Cell Culture

Original cell lines were commercially obtained. (RD cells: Cellosaurus accession No. CVCL_1649, ATCC No. CCL-136. RH30: Cellosaurus accession No. CVCL_0041, DSMZ ACC 489.) Throughout the study, GFP/Luciferase-expressing RD and RH30 cells were used, labeled as RH30/RD-GFP. Cells were generated as previously described [26]. Cell lines were cultured in a humidified CO2-enriched atmosphere in RPMI-1640 GlutaMAX-I (Life Technologies, Eggenstein, Germany) medium for RH30-GFP or DMEM GlutaMAX-I (Life Technologies) medium for RD-GFP, supplemented with 10% fetal calf serum (Life Technologies), 1% penicillin/streptavidin (Life Technologies) and 1 mM sodium pyruvate (Life Technologies). Routinely performed PCR mycoplasma testing displayed negative results (Minerva-Biolabs, Berlin, Germany).

### 2.3. NK Cell Enrichment from Peripheral Blood Mononuclear Cells (PBMCs)

PBMCs from healthy donors (buffy coats provided by DRK Blutspendedienst, Frankfurt, Germany) were enriched using density gradient centrifugation with Histopaque-1077 (Sigma-Aldrich, Taufkirchen, Germany). NK cells from PBMCs were enriched using a negative selection kit following the manufacturer’s instructions (EasySep™ Human NK Cell Enrichment Kit; STEMCELL Technologies, Cologne, Germany). NK cells were expanded and activated in NK-MACS (Miltenyi Biotec, Bergisch Gladbach, Germany) supplemented with human IL-15 (Peprotech, Rocky Hill, CT) for 15 days. The NK cell population purity was validated to be above 95% after enrichment (day 0), during expansion (day 7) and on the day of experiments (day 15) using a flow cytometric staining panel. NK cells were defined as CD56 and CD16 double-positive and CD3 negative.

### 2.4. Spheroid Generation and Co-Cultivation

Multicellular tumor spheroids were generated as previously described [23,27]. In brief, 5000 RD-GFP cells or 2500 RH30-GFP cells were seeded in a volume of 100 µL of complete culture medium per well using a 96-well ultra-low attachment (ULA) plate (Corning Inc., Corning, NY, USA). After seeding, the plate was centrifuged at 1000× *g* for 10 min at room temperature. After 3 days of spheroid growth, BV6 was added for 24 h to the wells, reaching a total volume of 200 µL per well. On day 4 post-seeding, 150 µL of medium was aspirated, and fresh medium containing expanded NK cells and inhibitors was added to the appropriate wells. The co-cultivation of RMS-GFP spheroids and NK cells was carried out for 5 days. The assessment of GFP fluorescence was performed using an ImageXpress XLS Widefield analysis system (Molecular Devices, Sunnydale, CA, USA). To this end, the best projection image of a 25-layer z-stack was stored and analyzed using MetaXpress (Molecular Devices; v. 6.5.4.532) and FIJI (ImageJ v. 1.53c [28]). On day 5, dead cells were stained with 2 µg/mL propidium iodide (PI; Sigma-Aldrich). As an indicator of spheroid killing, the PI-to-GFP fluorescence ratio, normalized to the untreated control, was calculated.

### 2.5. Immunoblotting

Immunoblotting was performed using the following antibodies: mouse anti-cIAP1 (Santa Cruz, Heidelberg, Germany; 271418), rabbit anti-cIAP2 (Cell Signaling Technologies, Leiden, The Netherlands; 3130), rabbit anti-survivin (R&D; AF886), mouse anti-XIAP (BD, Franklin Lakes, NJ, USA; 610716), rabbit anti-IκBα (Cell Signaling Technologies; 9242), mouse anti-p-IκBα (Cell Signaling Technologies; 6249L), mouse anit-p65 (Santa Cruz; sc-8008), rabbit anti-p-p65 (Cell Signaling Technologies; 3033S), rabbit anti-NIK (Cell Signaling Technologies; 4994), mouse anti-p100 (Merck-Millipore, Darmstadt, Germany; 05-361), mouse anti-GAPDH (BioTrend, Köln, Germany; NB-29-00852), mouse anti-vinculin (Sigma/Merck; V9131), mouse anti-caspase-8 (Enzo; ADI-AAM-118-E), rabbit anti-caspase-9 (Cell Signaling Technologies; 9502S), rabbit anti-DR5 (Merck-Millipore; AB16942) and mouse anti-PARP-1 (Cell Signaling Technologies; 9546S). The following secondary horseradish peroxidase-conjugated antibodies were used: goat anti-mouse (Abcam, Berlin, Germany; ab6789) and goat anti-rabbit (Abcam; ab6721). The quantification of Western blot and the area determination of the region of interest were performed using ImageJ/FIJI [28]. Subsequent ratio calculations were performed using Microsoft Excel.

### 2.6. qRT-PCR

RNA was isolated using a peqGOLD MicroSpin total RNA kit and a peqGOLD total RNA kit according to the manufacturer’s instructions, including DNase I digestion (PeqLab, Erlangen, Germany). In total, 1 µg of mRNA was used for cDNA synthesis using a RevertAid first-strand cDNA synthesis kit (ThermoFisher, Roskilde, Denmark). qRT-PCR was performed using a QuantStudio™ 7 Flex system (Applied Biosystems, Darmstadt, Germany) using Sybr™ Green PCR master mix (Applied Biosystems, Darmstadt, Germany) and the following primers: cIAP1 (fwd, GATATTGTGTCACGACTTCTTAATGC; rev, TCTGTTCTTCCGAATTAATGACAA), cIAP2 (fwd, GATGAAAATGCAGAGTCATCAATTA; rev, CATGATTGCATCTTCTGAATGG), CCL5 (fwd, CGCTGTCATCCTCATTGCTA; rev, GAGCACTTGCCACTGGTGTA), MMP9 (fwd, CTTTGAGTCCGGTGGACGAT; rev, TCGCCAGTACTTCCCATCCT), IL-8 (fwd, CTCTTGGCAGCCTTCCTGATT; rev, TATGCACTGACATCTAAGTTCTTTAGCA), DR5 (fwd, AGACCCTTGTGCTCGTTGTC; rev, TTGTTGGGTGATCAGAGCAG), G6PD (fwd, ATCGACCACTACCTGGGCAA; rev, TTCTGCATCACGTCCCGGA), RPII (fwd, GCACCACGTCCAATGACAT; rev, GTGCGGCTGCTTCCATAA), 18S-rRNA (fwd, CGCAAATTACCCACTCCCG; rev, TTCCAATTACAGGGCCTCGAA) and 28S-rRNA (fwd, TTGAAAATCCGGGGGAGAG; rev, ACATTGTTCCAACATGCCAG).

### 2.7. RNA Sequencing

mRNA-focused RNA libraries were prepared using an Illumina TruSeq stranded RNA library prep kit with input amounts of 200 or 400 ng of total RNA. Poly(A)-positive mRNA transcripts were isolated from total RNA using binding to magnetic oligo(d)T beads and were subsequently fragmented and reverse-transcribed during first- and second-strand synthesis to yield double-stranded cDNA. The fragmented cDNA was then end-repaired and adenylated, followed by the ligation of the appropriate eight-nucleotide NEXTFLEX DNA barcode adapter (PerkinElmer). The final ligation product was amplified using PCR. Intermediate and final library purification steps were performed using AMPure XP beads (Beckman Coulter). Final library quantity and quality were assessed using Qubit dsDNA HS AssayKit (Invitrogen) and an Agilent High Sensitivity DNA kit. All libraries were sequenced in paired-end mode (2 × 50 nt) using an Illumina NovaSeq 6000 S2 flow cell, resulting in ~30 million distinct sequencing reads per library. Sequencing resulted in 18 sequencing-read libraries containing between 26.1 million and 37.5 million sequencing reads (mean of 31.1 million). The reads were mapped against the human genome (build GRCh38.95) using STAR (version 2.6.1d). We obtained gene counts using a custom union model of isoforms of transcripts per gene based on the ENSEMBL (GRCh38.95) known gene database. Gene counts were normalized using the DESeq2 median of ratios. Processed data are available at GEO (accession number GSE223787). The differential expression analyses of groups were performed using the Wald test implemented in DESeq2. For visualization purposes, the heatmap of differentially expressed genes (DEGs) was created using the shifted logarithm-transformed expression values. The identified DEGs were post hoc-analyzed using the web-based software platform Metascape (https://metascape.org, accessed on 29 September 2020), including analysis with TRRUST [29,30].

### 2.8. siRNA Transfection of RMS-GFP Spheroids

Transfection with siRNA (Silencer^®^ select; ThermoFisher) followed a reverse protocol for the lipofectamine RNAiMAX transfection reagent protocol (Life Technologies). The following knockdown assays were used for caspase-9: No. 1, s2428; No. 2, s2429; No. 3, s2430. For caspase-8, we used the following: No. 1, s2427; No. 2, s2425; No. 3, s2426. For NIK, we used the following: No. 1, s17187; No. 2, s17186; No. 3, s17188. Spheroids were allowed to grow for 3 days before the addition of treatment or harvesting to control for successful target knockdown.

### 2.9. Statistical Analysis

Statistical analysis was performed using GraphPad Prim v9 using Student’s t-test and 2-way ANOVA, followed by post hoc analysis, if needed. Asterisks indicate significance levels, i.e., n.s., ≥ 0.05; *, *p* < 0.05; **, *p* < 0.01; ***, *p* < 0.001. If not stated otherwise, data are depicted as means +/− SEM.

## 3. Results

### 3.1. BV6 Facilitated the Increased NK Cell Killing of RMS-GFP Spheroids

To model an in vivo-like cell state of solid tumor cell lines, we used a scaffold-free multicellular tumor spheroid system as previously described [23,27,31]. GFP-expressing RD and RH30 cell lines were able to form spheroids within 3 days of culture, as indicated by increased GFP intensity and higher compactness (Figure 1). Pre-treatment with BV6 did not significantly change the growth behavior of RD or RH30 spheroids, confirming that BV6 acts as a cell death-sensitizing agent rather than inducing cell death on its own. Co-cultivation with cytotoxic IL-15-activated NK cells in an effector-to-target ratio (E:T) of 1:1 could control the growth of untreated RD spheroids, but not that of untreated RH30 spheroids. Here, RH30 spheroids lost their compact phenotype but continued to grow. The combination of BV6 pre-treatment and co-cultivation with NK cells could significantly decrease the growth of both RD and RH30 spheroids, depicted as decreased GFP intensity and size of spheroids (Figure 1). These data demonstrate that the addition of BV6 facilitates and increases NK cell attack. Of note, in a higher E:T ratio, 5:1, NK cells were able to completely eliminate the spheroids (Appendix A). In a co-cultivation killing assay of suspended RMS and NK cells, BV6 also revealed a sensitizing effect (Appendix A) independently of the culture model. To exclude that BV6 had an unwanted side effect on NK cells, we investigated the effect of BV6 on activated NK cells. Treatment with BV6 alone at 2.5 and 5 µM did not induce significant levels of cell death in IL-15-activated NK cells (Appendix A). In addition to BV6, we used four different Smac mimetics (ASTX660, GDC-0152, LCL161 and AZD5582) to assess if the observed BV6-induced sensitization is a general mechanism of Smac mimetics (Appendix A). All combinations showed non-significant effects compared with the respective control. However, a slight trend might be visible in AZD5582-treated RD-GFP spheroids in E:T ratios of 0.5:1 and 5:1.

### 3.2. Activation of NF-κB Signaling Pathways and Transcriptome Regulation by BV6

To analyze the underlying molecular mechanisms of sensitization induced by BV6 pre-treatment, we initially focused on the alterations in NF-κB signaling and downstream transcriptional changes mediated by BV6 [32,33]. As expected, BV6 treatment induced rapid degradation of cIAP1/2, in both 2D and 3D spheroid cultures. XIAP degradation could only be observed in 3D spheroids (RH30 (Figure 2A,B); RD (Appendix A)). Following the degradation of cIAP proteins, both the canonical and non-canonical NF-κB signaling pathways were activated. Slightly increased IκBα phosphorylation was detectable after 1 h of BV6 treatment, followed by increased p65 phosphorylation. At the same time, the partial degradation of p100 to transcription factor p52 was detectable, accompanied by the accumulation of NF-κB-inducing kinase (NIK). The activation of both NF-κB pathways was also visible in 3D spheroids after 24 h of BV6 treatment (RH30 (Figure 2C,D); RD (Appendix A)).

To further investigate the NF-κB signaling pathway and the downstream transcriptional changes, we performed an RNAseq analysis of RH30 cells treated with BV6 (5 µM, 24 h). Figure 3A,B depict the volcano plot showing a total of 182 DEGs and the top 50 regulated genes upon BV6 treatment. The analysis of putative responsible transcription factors using TRRUST (Figure 3C) revealed strong involvement of p65 (RELA) and p105 (NFKB1), which are members of the canonical NF-κB pathway. Further, enriched pathway analysis showed the up-regulation of the NF-κB signaling pathway and, as the highest hit, the up-regulation of the innate immune response (Figure 3D,E), thus validating the results of the Western blot showing strong NF-κB activation by BV6. Further, multiple DEGs identified using RNAseq were validated with Fluidigm^®^ (Appendix A), showing a positive correlation of the log2 fold change values calculated with RNAseq and Fluidigm analyses. The impact of BV6 on transcriptional activation was validated in RD cells using qRT-PCR (Appendix A), showing that similarly to RH30 cells, also in RD cells, NF-κB target genes were transcriptionally up-regulated by BV6.

We further investigated the chemokine CCL5, as it was among the top 50 BV6-induced DEGs and has been described in the context of NK cell cytotoxicity and migration [34,35]. Firstly, we investigated whether CCL5 up-regulation was shown to be NIK-dependent by performing NIK knockdown. CCL5 up-regulation was NIK-dependent and led to increased secretion as well as to an increased storage in intracellular granules (Appendix A–D). Secondly, to analyze the putative effect of NK cell killing sensitization, we performed the CRISPR/Cas9 knockout of CCL5 in RH30 cells, which did not alter the killing activity in our RH30 spheroid NK cell co-culture model. Therefore, we conclude that although being a prominent transcriptional target of BV6, CCL5 is not essential to mediating the sensitizing effect induced by BV6 (Appendix A).

### 3.3. NIK as Transcriptional Master Regulator of BV6-Induced Transcriptional Change

As NIK showed increased accumulation upon BV6 treatment and the transcriptional regulation of our top hit CCL5 was NIK-dependent, we further investigated the role of NIK in the observed effect of BV6 sensitization to increased NK cell attack. The siRNA-mediated knockdown of NIK in RH30 cells resulted in a decreased activation of both the non-canonical and canonical NF-κB signaling pathways, visualized as decreased partial degradation of p100 to p52 and reduction in p-p65 (Figure 4A,B). The silencing of NIK led to a reduction in NF-κB target gene transcription, i.e., cIAP1, cIAP2, IL-8, MMP9 and DR5, in RH30 (Figure 4C,D; Appendix A) or RD cells (Appendix A). However, the sensitizing effect of BV6 on NK cell-mediated RMS spheroid killing was not altered by NIK knockdown in either RH30 or RD spheroids (Figure 4E,F, Appendix A). Taken together, our results indicate that the BV6-mediated sensitizing effect may occur independently of the NIK-mediated transcriptional effects exerted by activated NF-κB signaling.

### 3.4. Investigation of NK Cell Killing Pathways

As the BV6-induced sensitization appeared to be independent of BV6-induced transcriptional changes, we focused on the intracellular cell death pathways induced in RMS spheroids upon NK cell attack. NK cells can kill their target cells by inducing extrinsic apoptosis. The characterization of the death receptors on RMS cells revealed that RD cells did not express Fas or TNFR1, but they were slightly positive for DR4 and highly positive for DR5, the receptors for TRAIL. On the other hand, RH30 cells were negative for Fas and DR4, but they presented TNFR1 and DR5 on their cell surface (Appendix A). Therefore, we first focused on TRAIL-mediated killing, as DR5 expression is shared between RD and RH30 cells. Blocking TRAIL with a TRAIL-targeting antibody (antiTRAIL) did not show a reduction in NK cell killing in BV6-pre-treated RD spheroids (Figure 5A). However, a slight but not significant reduction in killing could be observed in BV6-pre-treated RH30 spheroids co-cultured with NK cells (Figure 5B).

The functional effect of the TRAIL-blocking antibody was validated with co-treatment with recombinant TRAIL as the death-inducing compound. Here, the neutralizing antibody rescued the RMS spheroids from TRAIL-mediated cell death, confirming the on-target activity of the blocking antibody (Appendix A). 

As previously shown, BV6 activates the NF-κB pathway and has been described to induce a TNFα autocrine feedback loop [36]. To study whether the effect of BV6 and NK cells is TNFα dependent, we added the blocking antibody Enbrel. We added Enbrel as a co-treatment with BV6 and continued the co-treatment during NK cell co-cultivation. Although not statistically significant, we observed a trend in the kinetics during NK cell co-cultivation (Appendix A). PI end-point staining on day 5 showed a slight reduction in cell death in Enbrel-co-treated RD spheroids (Appendix A) but not in RH30 spheroids (Appendix A). Therefore, we conclude that the partial involvement of TNFα during NK cell-mediated killing may be possible.

### 3.5. Dissection of Caspase Involvement in NK Cell-Dependent Killing

Cell death mainly relies on the activation of the apoptotic caspase cascade, since the addition of the pan-caspase inhibitor zVAD.fmk could significantly reduce the NK cell-mediated killing of BV6-pre-treated RMS spheroids (Figure 5A,B). Western blot analysis revealed high cleavage of the executioner caspase-3 with the combination of BV6 and NK cells in RD and RH30 spheroids (Figure 5C,D). To investigate whether the extrinsic or intrinsic apoptotic pathway drives the caspase cascade upon BV6/NK cell co-treatment, we silenced the main initiators, caspases-8 and -9. Caspase-9 is involved in mitochondria-mediated apoptotic induction. Upon mitochondrial outer membrane permeabilization, the formation of the apoptosome occurs, and caspase-9 is activated [37]. The knockdown of caspase-9 did not reduce NK cell-mediated killing, despite the efficient loss of protein (Figure 6A,B), suggesting that mitochondrial involvement and the intrinsic apoptotic pathway are not required for the NK cell-mediated killing of RMS spheroids.

Upon extrinsic apoptotic induction, caspase-8 is activated in the DISC, leading to the activation of the caspase cascade, resulting in the execution of apoptotic cell death [37]. The knockdown of caspase-8 should, therefore, limit extrinsic cell death induction and thus may decrease the NK cell killing of RMS spheroids. While caspase-8 knockdown in RH30 spheroids did not reduce NK cell killing, a significant reduction in the BV6- and NK cell-mediated killing of RD spheroids was observed with siRNA construct No. 3, which also exhibited the best knockdown efficacy (Figure 6C–E). Taken together, these data indicate that in RD cells, caspase-8 and the extrinsic apoptotic pathway may play important roles in mediating BV6- and NK cell-triggered killing.

## 4. Discussion

In this study, we describe a novel way to utilize the apoptosis-sensitizing ability of Smac mimetics to increase the NK cell-mediated killing of pediatric tumor spheroids. Therefore, we hypothesize that NK cells may represent a possible cellular immunotherapy with the potential to attack and eliminate pediatric cancers such as RMS. NK cells and immune cells in general were only present in a minority of histological slices of RMS cases, indicating an immunologically cold/immunosuppressive TME [8,9]. Until now, only four different Smac mimetics (Debio 1143, ASTX660, APG-1387 and Birinapant) have reached clinical trials, of which Debio 1143 is currently the only one being tested in phase 3. At the moment, Smac mimetics are still investigational drug compounds and have not achieved clinical approval. Here, we show that the cell death-sensitizing agent BV6, which is still under pre-clinical evaluation, might be a promising approach to counteract this immunosuppressive TME, as the BV6 pre-treatment of in vivo-like RH30 and RD multicellular spheroids could induce an effect of IL-15-activated NK cell-mediated killing sensitization. Among the Smac mimetics investigated here, only BV6 showed prominent sensitization to NK cell killing. This may be explained by an altered IAP recognition pattern due to different Smac mimetics. Further, the degradative efficacy of the individual compounds might be different, which leads to distinct molecular downstream mechanisms. Using BV6, Fischer et al. could identify a similar effect in an RMS:NK cell suspension co-culture model. However, the BV6 sensitizing effect was only present at high BV6 concentrations (10 µM) and at high E:T ratios (10:1) [24]. In contrast, we observed a sensitizing effect on RMS spheroids at lower concentrations of BV6 (5 µM), thus also improving the therapeutic window in which BV6 is not toxic to NK cells. Further, we could lower the effective E:T ratio to 1:1 as an approach to remodel an environment little infiltrated by immune cells. Of note, this BV6-induced sensitizing effect on RMS cells is also present in co-cultivation with cytokine-induced killer (CIK) cells [25], suggesting an underlying tumor-centric molecular mechanism.

A tumor-centric approach to unravel the molecular mechanism driving BV6-induced sensitization is to identify direct BV6 targets and further downstream pathways. As a Smac mimetic, i.e., IAP antagonist, the initial molecular targets of BV6 are known and could be reproduced here in RMS cells and spheroids. Upon treatment, BV6 induces rapid cIAP1/2 degradation, as previously observed in leukemia [38], colorectal cancer [39] and lung cancer cells [40]. Additionally, these studies could show that BV6 was not able to degrade XIAP in monolayer cell cultures, which is in line with our observation of exclusive XIAP degradation in RMS spheroids, but not in RMS monolayer cell cultures. The importance of XIAP in a spheroidal system was proven by Gallardo-Pérez et al., who showed that the loss of XIAP could re-sensitize breast cancer, head and neck cancer, and cervical cancer spheroids to cytotoxic agents and also reduce spheroidal volume [41].

BV6-induced cIAP1/2 degradation blocks the poly-ubiquitylation of NIK. Thus, NIK accumulates in the cytoplasm and leads to the downstream activation of the non-canonical NF-κB signaling pathway in RMS spheroids and monolayer cell cultures. In both spheroids and monolayer cell cultures, we observed a BV6-mediated activation of both the canonical (phosphorylation of p65) and non-canonical (p100 partial degradation to p52) NF-κB signal transduction pathways. This NF-κB activation leads to the broad transcriptional remodeling of RH30 cells, as identified using our exploratory RNAseq approach. Our transcriptomic results are in agreement with several genes of interest identified in previous microarrays analyzing BV6-mediated transcriptional regulation in breast cancer cells [32] and chronic lymphocytic leukemia [42]. Further, pathway enrichment analysis validated the up-regulation and involvement of NF-κB signaling pathways in BV6-treated RH30 cells. One gene of interest that was increased upon BV6 treatment in RH30 cells was the chemokine CCL5, which was formerly described as being an NK cell-activating [34] and pro-migratory compound [43]. BV6-induced CCL5 expression and secretion (Appendix A) led to the assumption of a CCL5-NK cell axis, which putatively facilitates BV6-induced sensitization. However, using CRISPR/Cas9-mediated CCL5 knockout in RH30 spheroids, we observed that CCL5 was not required for BV6-induced sensitization to NK cell attack.

Most of the transcriptional changes observed upon BV6 treatment, including the up-regulation of CCL5, were mediated by NIK. As NIK is directly affected by the degradation of cIAP1/2 proteins, we hypothesized an involvement of NIK and of the non-canonical NF-kB pathway in the BV6-induced effect of NK cell-mediated killing sensitization. A similar involvement of NIK could be observed in MDA-MB-231 breast cancer cells, where siRNA-mediated NIK knockdown reduced the BV6-mediated transcription of several target genes as well as apoptotic cell death induction [33]. In the same publication, BV6-mediated TNFα autocrine stimulation led to RIPK1-dependent cell death [33]. However, the successful siRNA-mediated NIK knockdown in our spheroid model did not reduce BV6/NK cell-mediated killing, indicating an NIK-independent and putatively NF-κB-independent mechanism that drives BV6-induced sensitization.

The second molecular consequence of BV6-induced IAP degradation is the putative release and activation of caspases. In RMS spheroids, pre-treatment and co-cultivation with NK cells led to increased activity of the executioner caspase-3 (Figure 5C). A general involvement of caspases in our RMS spheroid model could be established with co-treatment with the pan-caspase inhibitor zVAD.fmk, which shows a decreased NK cell-mediated killing and BV6-induced sensitizing effect. A similar effect was observed in hepatocellular carcinoma cells in co-cultivation with NK cells and co-treatment with the Smac mimetic APG-1387. The increased killing was correlated with elevated caspase-3 activity upon combination of APG-1387 and NK cell co-cultivation. Here, the authors proposed a caspase- and RIPK1-dependent mechanism of APG-1387-induced sensitization [44].

Further, our RMS spheroids model illustrated TRAIL independence of NK cell-mediated killing, as co-treatment with a TRAIL-neutralizing antibody did not rescue RMS spheroids from BV6-sensitized NK cell-mediated killing. This is in contrast to a previous study, where the partially TRAIL-dependent NK cell killing of BV6-stimulated RMS cells could be observed [24]. This difference might be explained by an altered killing capacity of NK cells in different culture models. Since RMS tumors show an immunosuppressive TME [8,9], our 3D RMS spheroids may display an in vivo-like suppressive phenotype. This may modify the killing mechanism of NK cells into a more perforin/granzyme B-dependent mode of killing.

The next step to decipher the apoptotic signaling induced by NK cell attack was the identification of the relevant caspase initiator responsible for the activation of the executioner caspase-3 [45]. Two recent publications highlighted the importance of the mitochondrial pathway during NK cell-mediated tumor cell killing [23,46]. As caspase-9 is the apical caspase in the intrinsic, mitochondrial apoptosis pathway, we performed caspase-9 siRNA-knockdown studies in our RMS spheroid model. However, the BV6-induced sensitized NK cell killing of RMS spheroids occurred independently of caspase-9. 

The other main apoptotic pathway involves the ligation of death receptors and the intracellular assembly of the DISC, leading to the activation of caspase-8 as the apical initiator caspase [13]. It is known that NK cells present death ligands on their surface and, with a serial killer mechanism, are able to switch from granzyme B-mediated to death ligand-caspase-8-dependent killing [12]. As previously identified, caspases in general were involved, but independently of TRAIL, in our RMS spheroid model of BV6/NK cell-induced killing. A potential role of caspase-8 in the NK cell killing of RMS spheroids was identified upon siRNA-mediated caspase-8 knockdown. This knockdown led to an RMS cell type-dependent involvement of caspase-8, since the knockdown of caspase-8 in RD spheroids was sufficient to reduce the BV6-sensitized NK cell-mediated killing. However, the knockdown of caspase-8 did not rescue RH30 spheroids from BV6-sensitized NK cell attack. In contrast to RD cells, RH30 cells also express caspase-10 [47,48], suggesting that upon the removal of caspase-8, caspase-10 may take over as the apical apoptotic initiator caspase and induce RIPK1-dependent cell death [49].

## 5. Conclusions

Taken together, BV6 leads to the activation of NF-κB-dependent transcriptional regulation, which is possibly not responsible for the observed pro-NK cell sensitizing effect. In the cell death machinery, a putative role of caspases, especially caspase-8, might be relevant as cell death mediator. However, this mechanism is mainly independent of death receptors and TNFα. Such signaling events could be described as the pre-assembly of a complex comprising RIPK1, caspase-8 and FADD, termed ripoptosome [50,51]. By using an in vivo-like RMS spheroid model, we could further disentangle the molecular mechanisms of the Smac mimetic BV6 in conjunction with cytolytic NK cells. Inspired by the Nobel prize awarded to G.B. Elion and G.H. Hitchings in 1988 and the fundamental paradigm shift in drug development [15], we applied a specific IAP protein antagonist to increase the killing effect of NK cells on RMS cells and spheroids. Therefore, our study not only proposes a cancer cell-centric approach for specifically targeting IAP proteins, but by combining players of the cellular innate immune system, it shows a promising strategy to increase the cytolytic effect of NK cells, as an alternative strategy to the 2018 Nobel prize awarded for the discovery of immune checkpoint proteins [7].

## Figures and Tables

**Figure 1 cells-12-00906-f001:**
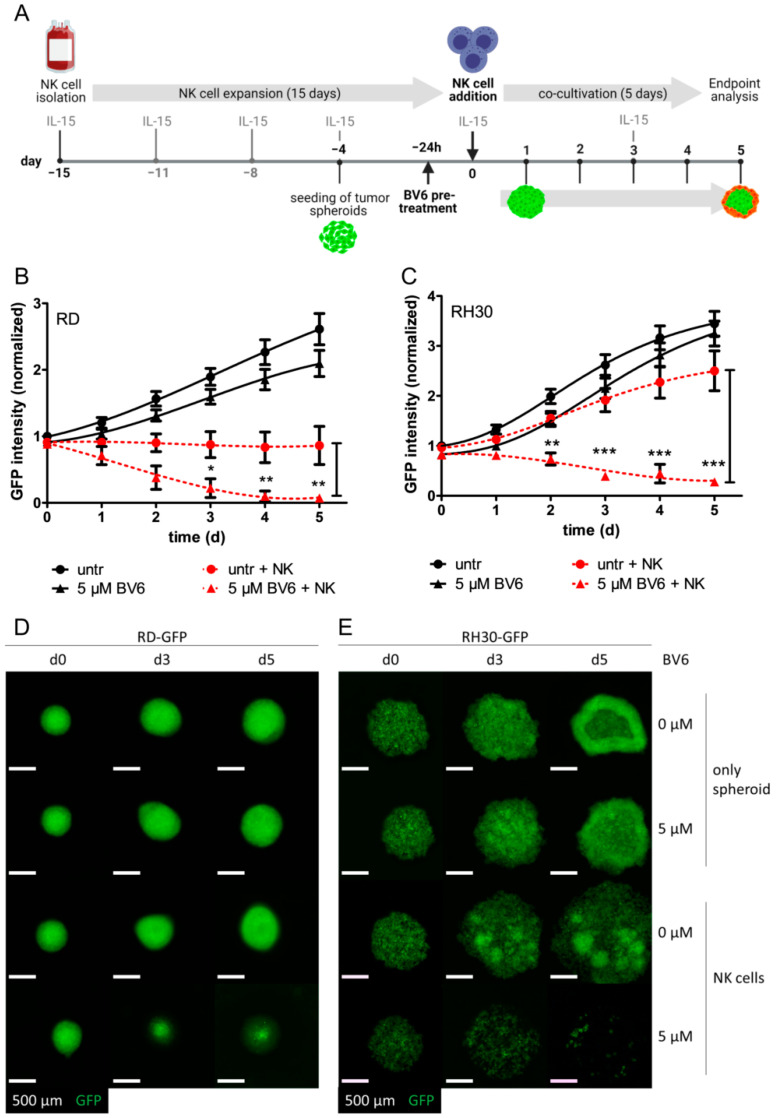
The BV6-sensitized NK cell-mediated killing of RMS spheroids. (**A**) Experimental setup of the RMS spheroid-NK cell experiment. Subsequent to NK cell enrichment using primary PBMCs, cells were expanded and activated with IL-15 for 15 days before being used in co-culture experiments. RMS spheroids were seeded four days prior to NK cell co-cultivation. After 3 days, spheroids were pre-treated with BV6 for 24 h. On the day of co-cultivation, BV6 was washed out, and NK cells were added with fresh IL-15, followed by co-cultivation for 5 additional days, during which microscopic images were obtained each day. Killing curves of BV6-treated or untreated RD (**B**) or RH30 (**C**) spheroids in co-cultivation with NK cells (E:T ratio of 1:1) or as tumor spheroids only. GFP intensities were normalized to the start of co-cultivation (untreated day 0). N = 5, mean +/− SEM; statistical analysis was performed using two-way ANOVA with Bonferroni multiple comparison test. *, *p* < 0.05; **, *p* < 0.01; ***, *p* < 0.001. Microscopic image of GFP-fluorescent RD (**D**) or RH30 (**E**) spheroids upon BV6 stimulation and NK cell co-cultivation at selected time points; scale bar equal to 500 µm.

**Figure 2 cells-12-00906-f002:**
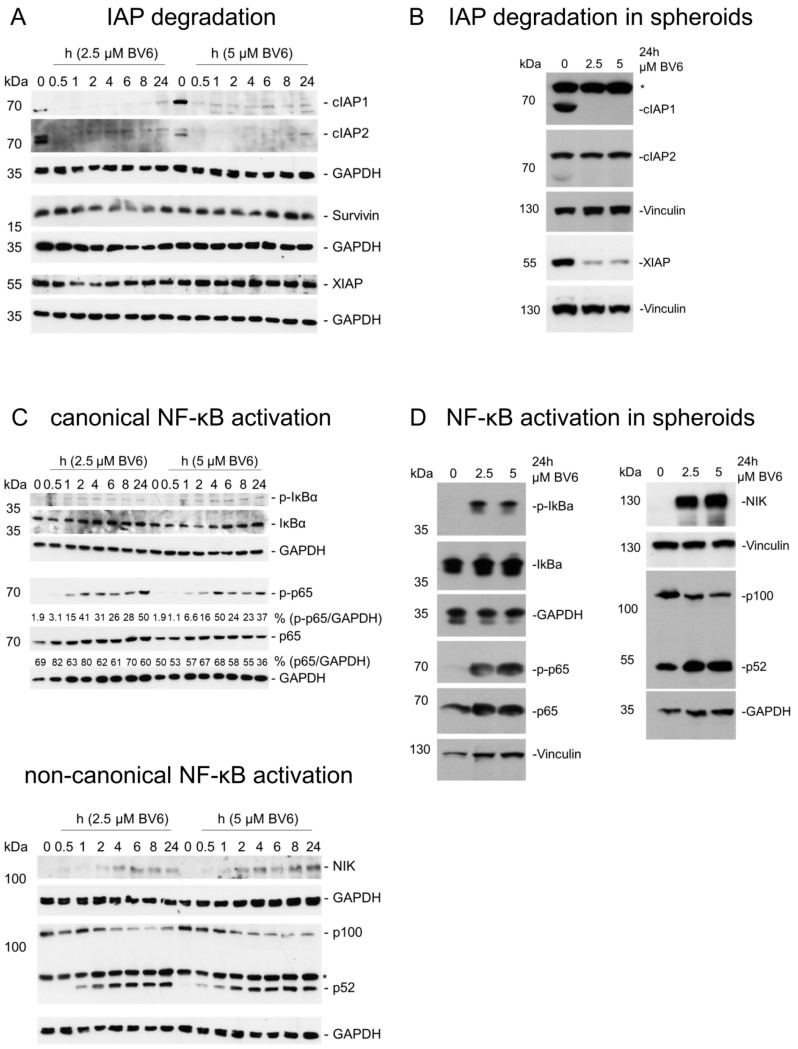
Validation of NF-κB activation in RH30 cells and spheroids. Western blot analysis showing BV6-induced changes in IAP protein levels in RH30 cells cultured in monolayer cultures (**A**) or as spheroids (**B**). BV6-regulated canonical and non-canonical NF-κB signaling pathways in RH30 cells cultured in monolayer cultures (**C**) or as spheroids (**D**). Western blots are exemplary of at least two independent experiments. If applicable, Western blot quantification is depicted as fold changes of GAPDH-normalized target protein expression using the untreated condition as reference. * depicts unspecific signal.

**Figure 3 cells-12-00906-f003:**
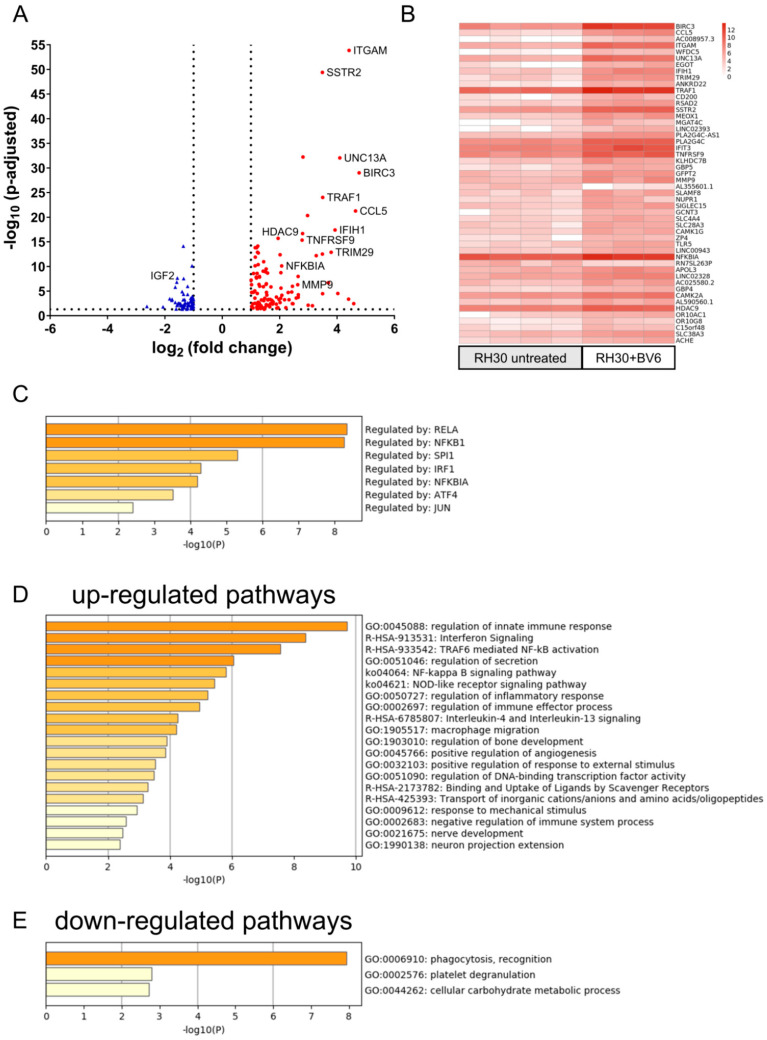
RNAseq analysis of transcriptional regulation by BV6. (**A**) Volcano plot of BV6-treated RH30 cells compared with untreated RH 30 cells, with down-regulated genes depicted in blue, up-regulated genes in red, and the 50 top regulated genes being labeled. (**B**) Log2 fold change values of the top 50 regulated genes depicted as heat map separately showing each biological replicate. (**C**) TRRUST analysis of putative involved transcription factors responsible for BV6-up-regulated genes. Pathway enrichment analysis of up- (**D**) or down-regulated genes (**E**) upon BV6 stimulation performed with web application metascape.org [29].

**Figure 4 cells-12-00906-f004:**
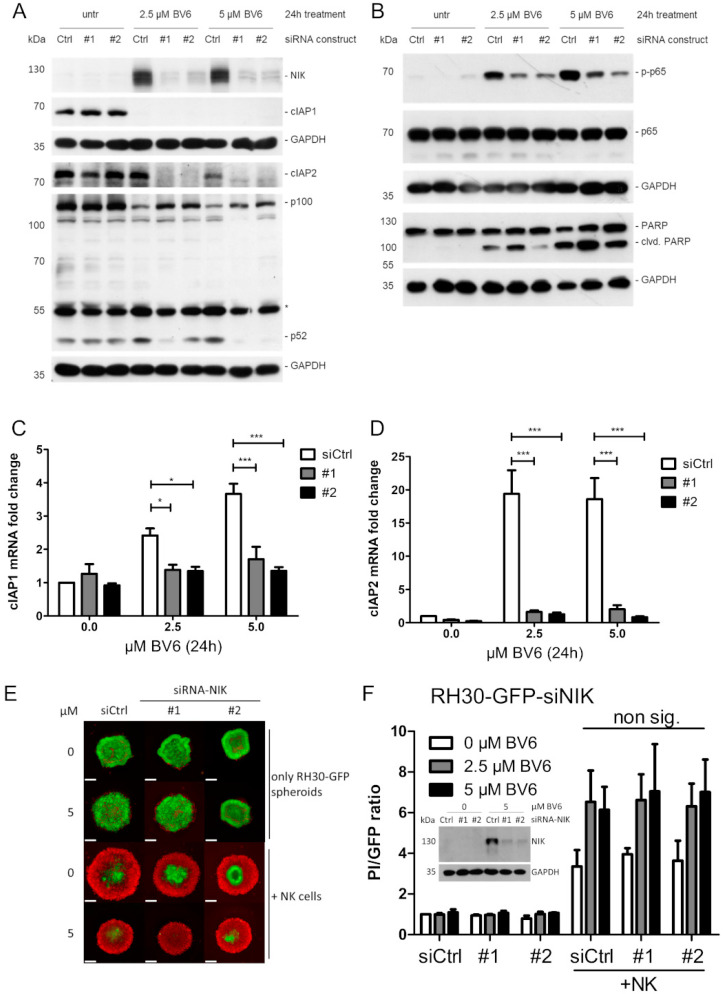
NIK as master regulator of BV6-induced transcription. Western blot analysis showing IAP degradation, and non-canonical (**A**) and canonical (**B**) NF-κB signaling pathways upon BV6 stimulation with siRNA-mediated NIK knockdown in RH30 cells. Western blot images are representative of at least two independent experiments. Relative mRNA quantification of NF-κB target genes (cIAP1 (**C**) and cIAP2 (**D**)) using qRT-PCR. N = 3, mean +/− SEM; statistical analysis conducted using two-way ANOVA with Tukey’s multiple comparison test. *; *p* < 0.05; ***, *p* < 0.001. RH30 spheroid generation upon siRNA NIK knockdown. (**E**) Microscopic images of BV6-stimulated GFP-fluorescent, NIK-KD RH30 spheroids, co-cultivated with NK cells and counter-stained with PI to detect dead cells; scale bar equal to 500 µm. (**F**) Quantification and PI/GFP ratio as read-out of induced cell death. N = 3, mean +/− SEM. Western blot insert depicts knockdown validation in RH30 spheroids.

**Figure 5 cells-12-00906-f005:**
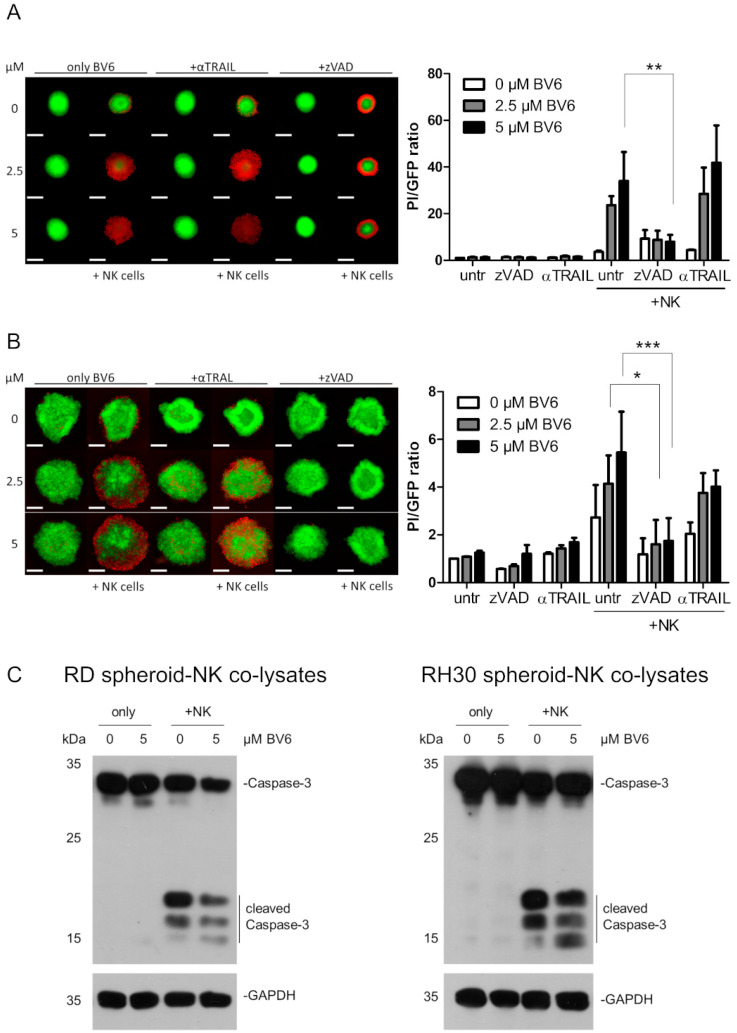
Involvement of TRAIL and caspases in the NK cell killing of RMS spheroids. RD (**A**) or RH30 (**B**) spheroids were pre-treated with BV6 at indicated concentrations, co-cultured with NK cells and either co-treated with the pan-caspase inhibitor zVAD.fmk (50 µM) or a TRAIL-neutralizing antibody (1 µg/mL). Quantification and PI/GFP ratio as cell death indicators. n = 3, mean + SEM; statistical analysis conducted using two-way ANOVA with Bonferroni multiple comparison test. *, *p* < 0.05; **, *p* < 0.01; ***, *p* < 0.001. Scale bar equal to 500 µm. (**C**) Western blot of cleaved, activated caspase-3 upon BV6 pre-treatment and NK cell co-cultivation with RD (left) and RH30 (right) spheroids. Images are representative of at least two independent experiments.

**Figure 6 cells-12-00906-f006:**
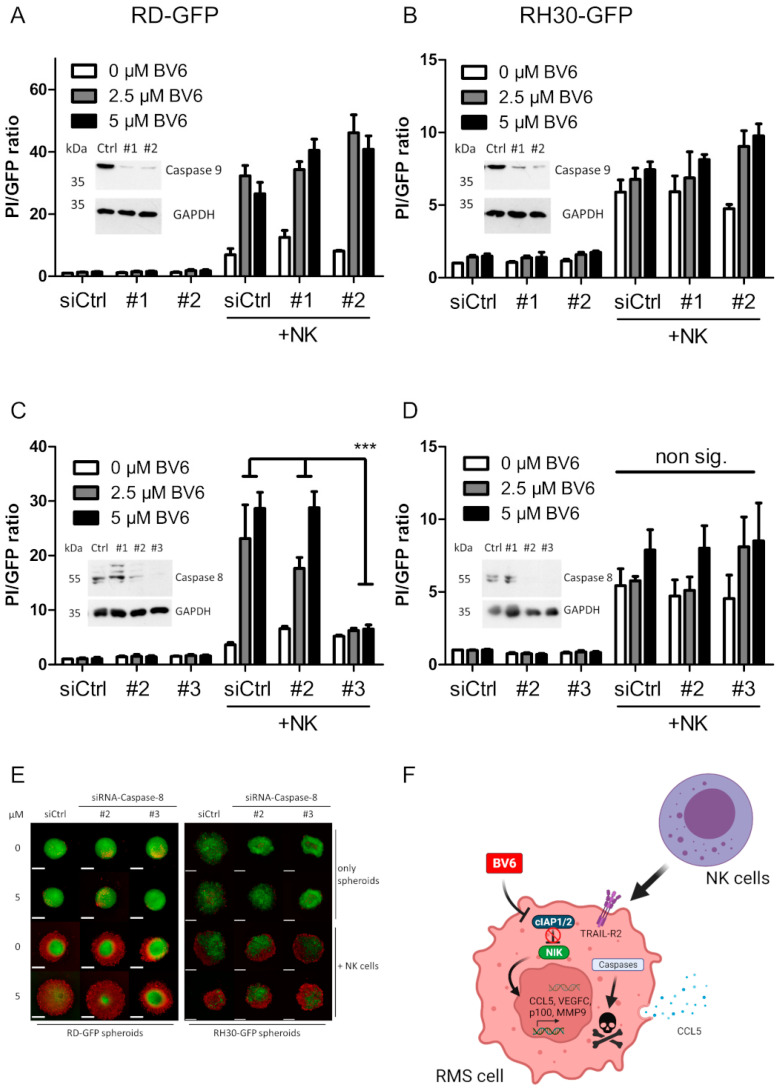
The NK cell killing of RMS spheroids is dependent on caspases. siRNA-mediated caspase-9 knockdown in RD (**A**) and RH30 (**B**), in addition to siRNA-mediated caspase-8 knockdown in RD (**C**) and RH30 (**D**) spheroids, followed by BV6 pre-treatment and NK cell co-cultivation. Quantification and PI/GFP ratio as cell death indicators. n = 3, mean + SEM; statistical analysis conducted using two-way ANOVA with Tukey’s multiple comparison test. ***, *p* < 0.001. (**E**) Microscopic images of GFP-fluorescent caspase-8-knockdown RH30 spheroids counterstained with PI; scale bar equal to 500 µm. (**F**) Graphical summary of BV6 putative molecular action in RMS cells and spheroids leading to sensitized NK cell killing.

## Data Availability

RNA sequencing data are available at GEO (https://www.ncbi.nlm.nih.gov/geo/, accessed and submitted on 14 January 2023) under accession number GSE223787.

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
