# Peer review of "Characterization of BV6-Induced Sensitization to the NK Cell Killing of Pediatric Rhabdomyosarcoma Spheroids"

_cells, 2023, doi:10.3390/cells12060906_

Round 1

Reviewer 1 Report

In this study the authors have investigated the underlying mechanism behind BV6-mediated sensitization of Rhabdomyosarcoma cell lines to activated NK cells. The study builds on previous investigations by the same group and is here extended by studying NK targeting of pre-sensitized spheroid cultures and extensive mapping of the dominant gene programs induced by BV6 in the spheroids. Although the study fails to identify a single responsible mechanism, the study is robust, and conclusions are supported by the data. Moreover, the negative finding suggesting that the pronounced activation of NFkB in the target cells is not responsible for the enhanced NK cell sensitivity has merits on its own. 

It would perhaps be interesting to briefly discuss the current status of smac-mimetics in the clinic and how the authors envisage that the reported synergy could be used. 

Can the authors speculate on why the other smac-mimetics fail to sensitize the target cells to NK cell killing? 

In contrast to the previous paper on this topic by the same group, TRAIL does not seem to be involved in the NK recognition of BV treated targets. Others have also reported that BV6 induces susceptibility to TRAIL-mediated killing. Can the authors comment on this discrepancy?

For the western blot experiments it would be good if the authors could perform a quantification of the data. The GADPH seems to increase with time in Figure 2C (canonical), which makes interpretation of the p-p65 somewhat difficult.

Typo. BV6-induced cIAP1/2 degradation blocks the poly-ubiquitylation of NIK.

Author Response

It would perhaps be interesting to briefly discuss the current status of smac-mimetics in the clinic and how the authors envisage that the reported synergy could be used. 

Response:  A discussion on the clinical development of the different Smac mimetics has been added to the text. 

Change in the manuscript: Lines 395-398.

Can the authors speculate on why the other smac-mimetics fail to sensitize the target cells to NK cell killing? 

Response:  We believe that differences in target recognition and degradation may explain the differences between different compounds, and have added this hypothesis to the discussion.

Change in the manuscript: Lines 402-406

In contrast to the previous paper on this topic by the same group, TRAIL does not seem to be involved in the NK recognition of BV treated targets. Others have also reported that BV6 induces susceptibility to TRAIL-mediated killing. Can the authors comment on this discrepancy?

Response:  One of the important aspects of this study is the use of spheroid models, which may alter the microenvironment and the accessibility of drugs or cellular therapy. We hypothesize that these alterations may lead to a different mechanism of NK cell killing, and have added this to the discussion.

Change in the manuscript: Lines 468-474

For the western blot experiments it would be good if the authors could perform a quantification of the data. The GADPH seems to increase with time in Figure 2C (canonical), which makes interpretation of the p-p65 somewhat difficult.

Response:  We thank the reviewer for this important point and for allowing us to further improve our manuscript. The quantification of the western blots has now been added to Figure 2C, and the data obtained should further strengthen our conclusions.

Change in the manuscript: Quantification in Fig 2C added below the corresponding blots.

Typo. BV6-induced cIAP1/2 degradation blocks the poly-ubiquitylation of NIK.

Response:  Thank you for spotting this typo.

Change in the manuscript: Line 427

Reviewer 2 Report

In their manuscript “Characterization of a BV6-induced sensitization towards an NK cell killing of pediatric rhabdomyosarcoma spheroids” the authors investigate the effect of several Smac inhibitors on rhabdomyosarcoma cell lines. Only BV6 together with NK cells leads to reduced growth and increased apoptosis of spheroids. Through a series of biochemical, transcriptomic and cell biologic assays the authors try to find out how BV6 sensitizes the tumor cells for NK cell killing and through which mechanisms NK cell exert their cytotoxicity. They rule out several processes and show that the extrinsic apoptotic pathway is involved. How NK cells trigger this pathway remains unclear.

Major points

Experiments should be performed at least three times and quantitative analyses normalized to housekeeping genes should be presented. In Fig 2C for example, GAPDH and p65 bands increase in parallel with p-p65.

Transcriptomic results of the effect of BV6 on RH30 cells were validated using RT-PCR. This should also be done for RD cells in order to corroborate the major alterations in another cell line, as has been done for most other assays. 

Possible reasons why of all the Smac inhibitors tested only BV6 sensitizes for NK cell killing and how NK cells exert their cytotoxicity should be discussed. 

Minor points

It seems that the numbering of suppl Fig. 8 and 9 is reversed. 

The purity of enriched NK cells should be documented.

Author Response

Major points

Experiments should be performed at least three times and quantitative analyses normalized to housekeeping genes should be presented. In Fig 2C for example, GAPDH and p65 bands increase in parallel with p-p65.

Response:  We thank the reviewer for pointing this out. The quantification of the western blots has now been added to Figure 2C, and the data obtained should further strengthen our conclusions.

Change in the manuscript: Quantification in Fig 2C added below the corresponding blots.

Transcriptomic results of the effect of BV6 on RH30 cells were validated using RT-PCR. This should also be done for RD cells in order to corroborate the major alterations in another cell line, as has been done for most other assays. 

Response:  To validate the transcriptional changes observed in the RH30 cells, we now show additional data derived from the RD cells in Supplementary Figure 3B.

Change in the manuscript: Addition of new Supplementary Figure 3B.

Possible reasons why of all the Smac inhibitors tested only BV6 sensitizes for NK cell killing and how NK cells exert their cytotoxicity should be discussed. 

Response:  We believe that differences in target recognition and degradation may explain the differences between different compounds, and have added this hypothesis to the discussion. The cytotoxic effect of NK cells is also discussed (lines 475-495).

Change in the manuscript: Lines 402-406

Minor points 

It seems that the numbering of suppl Fig. 8 and 9 is reversed. 

Response:  Thank you, this has been corrected.

Change in the manuscript: Correct order of Suppl Fig8/9.

The purity of enriched NK cells should be documented.

Response:  A section describing our assessment protocol for the purity of the isolated NK cells has been added.

Change in the manuscript: Lines 117-120.

Round 2

Reviewer 2 Report

The authors have sufficiently improved the manuscript.